# Challenges in access and satisfaction with reproductive, maternal, newborn and child health services in Nigeria during the COVID-19 pandemic: A cross-sectional survey

Mobolanle Balogun[1][*], Aduragbemi Banke-Thomas[2], Adekemi Sekoni[1], Godfred O. Boateng[3], Victoria Yesufu[1], Ololade Wright[4], Osinachi Ubani[5], Akin Abayomi[6], Bosede B. Afolabi[7], Folasade Ogunsola[8]

1 Department of Community Health and Primary Care, College of Medicine, University of Lagos, Idi-Araba, Lagos, Nigeria, 2 LSE Health, London School of Economics and Political Science, London, United Kingdom, 3 Department of Kinesiology, College of Nursing and Health Innovation, The University of Texas at Arlington, Arlington, Texas, United States of America, 4 Department of Community Health and Primary Health Care, Lagos State University College of Medicine, Ikeja, Lagos, Nigeria, 5 Lagos State Primary Health Care Board, Yaba, Lagos, Nigeria, 6 Lagos State Ministry of Health, Ikeja, Lagos, Nigeria, 7 Department of Obstetrics and Gynaecology, College of Medicine of the University of Lagos, Idi-Araba, Lagos, Nigeria, 8 Department of Medical Microbiology, College of Medicine of the University of Lagos, Idi-Araba, Lagos, Nigeria

☯ These authors contributed equally to this work.
* mbalogun@cmul.edu.ng

## Abstract

### Background

The presence of COVID-19 has led to the disruption of health systems globally, including essential reproductive, maternal, newborn and child health (RMNCH) services. This study aimed to assess the challenges faced by women who used RMNCH services in Nigeria's epicentre, their satisfaction with care received during the COVID-19 pandemic and the factors associated with their satisfaction.

### Methods

This cross-sectional survey was conducted in Lagos, southwest Nigeria among 1,241 women of reproductive age who had just received RMNCH services at one of twenty-two health facilities across the primary, secondary and tertiary tiers of health care. The respondents were selected via multi-stage sampling and face to face exit interviews were conducted by trained interviewers. Client satisfaction was assessed across four sub-scales: health care delivery, health facility, interpersonal aspects of care and access to services. Bivariate and multivariate analyses were used to assess the relationship between personal characteristics and client satisfaction.

### Results

About 43.51% of respondents had at least one challenge in accessing RMNCH services since the COVID-19 outbreak. Close to a third (31.91%) could not access service because

**Data Availability Statement:** All relevant data are within the manuscript and its Supporting information files.

**Funding:** The author(s) received no specific funding for this work.

**Competing interests:** The authors have declared that no competing interests exist.

they could not leave their houses during the lockdown and 18.13% could not access service because there was no transportation. The mean clients' satisfaction score among the respondents was 43.25 (SD: 6.28) out of a possible score of 57. Satisfaction scores for the interpersonal aspects of care were statistically significantly lower in the PHCs and general hospitals compared to teaching hospitals. Being over 30 years of age was significantly associated with an increased clients' satisfaction score (ß = 1.80, 95%CI: 1.10–2.50).

## Conclusion

The COVID-19 lockdown posed challenges to accessing RMNCH services for a significant proportion of women surveyed. Although overall satisfaction with care was fairly high, there is a need to provide tailored COVID-19 sensitive inter-personal care to clients at all levels of care.

## Introduction

Since 1990, there has been significant progress in reducing maternal mortality and morbidity globally [1]. However, in 2017, 284,000 women died of causes associated with pregnancy and childbirth, with almost all deaths (99%) occurring in low- and middle-income countries (LMICs). Nigeria accounts for 25% of the global maternal deaths [1]. Yet, the world was dealt a huge system shock in 2019—Coronavirus Disease (COVID-19). Following its declaration as a global pandemic by the World Health Organization (WHO) in March 2020, there has been an unprecedented disruption of health systems globally, including their capacity to provide health services as per usual [2, 3]. As of 5th April 2021, there have been an excess of 131 million confirmed cases, and almost 2.9 million deaths worldwide [3]. Specifically, as it relates to maternal heath, modelled estimates published during the earlier phase of the COVID-19 pandemic had already predicted that 8·3–38·6% increase in maternal deaths should be expected per month, as a result of direct and indirect causes [4].

Before the pandemic, Universal Health Coverage (UHC) had been set as a global goal to be achieved by 2030. This included emphasis by the WHO on the importance of women-centred care for mothers [5], recommending clear quality standards set for the sort of care expected at heath facilities for mothers and their newborns [6]. This goal required that health systems did not only focus on reducing the high burden of maternal deaths, stillbirths, and neonatal deaths that occur in LMICs, but also ensure that the needs of the women are met so that they are satisfied with the care that they receive [7]. Client satisfaction with care is a key component of quality of care [6], with evidence showing that poor satisfaction during facility-based births can have a negative effect on future use by affected women and other women within their sphere of influence [8, 9]. However, client satisfaction with care, including reproductive, maternal, newborn and child health (RMNCH) services, is multi-faceted and influenced by diverse factors [10–12].

The pandemic has been found to impact women disproportionately and one of the many ways is through access to health care. In particular, women in many LMICs have had challenges in accessing care with the resultant drop in antenatal care attendance and institutional deliveries in some of these countries [13]. There have also been additional stressors placed on skilled health personnel (nurses, midwives and doctors), large-scale lockdowns, restructuring of health services, increased cost of service utilization, and limited access to medical supplies

[14–16]. These occurrences have resulted in a pressing need to understand the level of satisfaction of women with experienced care during the COVID-19 pandemic as well as the determinants of satisfaction. Consequently, this study aimed at understanding the challenges faced by women who used RMNCH services in Lagos, Nigeria, their satisfaction with care received during the COVID-19 pandemic and the factors associated with their satisfaction.

## Methods

### Study design, site and population

This was a descriptive, cross-sectional survey conducted in Lagos, southwest Nigeria during the COVID-19 outbreak. Lagos State is divided into five administrative divisions namely Ikeja, Badagry, Ikorodu, Lagos and Epe. Health care delivery is structured along a three-tier system– tertiary, secondary and primary health care centres. The state has three public tertiary facilities which provide RMNCH services–Lagos University Teaching Hospital (LUTH), Lagos State University Teaching Hospital (LASUTH) and Federal Medical Centre, Ebute Metta. The state also has 26 secondary facilities (general hospitals) and 329 functional primary health care (PHC) facilities spread across the five administrative divisions, all of which provide RMNCH services i.e. integrated services for mothers and children from pre-pregnancy to delivery, the immediate postnatal period, and childhood. The RMNCH services provided vary according to the level of care and include; 1) Clinical care: case management for sexually transmitted infections, post-abortion care, skilled obstetric care at birth and essential care for neonates, prevention of mother to child transmission of HIV, emergency obstetric care and immediate emergency care for newborn babies, case management of childhood and neonatal illness, care of children with HIV; 2) Outpatient and outreach services: family planning, prevention and management of sexually transmitted infections and HIV, antenatal care, postnatal care, childhood vaccination, nutrition and growth monitoring.

The study population consisted of women of reproductive age (15–49 years) who had just received RMNCH services at one of the health facilities across the three tiers of care. Women were assessed to be eligible if they had accessed RMNCH services in the facility at least once during the COVID-19 outbreak in Lagos between 16th September 2020 and 12th October 2020. Those less than 18 years old were only included if they were emancipated, defined in this study as being married or living independently of parents.

### Sample estimation and selection

The minimum sample size of 400 was calculated using the Cochran's formula and was based on 5% margin of error, 95% confidence interval, proportion (p) of 62.5% which represents overall satisfaction with MCH services in a hospital in southwest Nigeria [17], and 10% addition for non-response and recording errors. This sample size was tripled to account for design effect in using multiple sites making a sample size of 1,200 women. There was an equal allocation of the sample size to the three levels of care (i.e., 400 women per level of care). This was further allocated equally across the facilities for each level of care.

Respondents were selected using a multi-stage sampling technique. At the first stage (facility level), the two teaching hospitals providing RMNCH services in Lagos State (LUTH and LASUTH) were purposively selected. Ten secondary facilities (two from each of the five administrative divisions) were selected by simple random sampling. Ten PHCs (two with the highest number of clients from each of the five administrative divisions) were selected purposively. At the participants level, consenting eligible participants were selected consecutively across all outpatient clinics providing RMNCH services until the sample size was attained.

## Data collection

Exit interviews were conducted with eligible end-users of RMNCH at the selected facilities. A pretested, structured questionnaire was used by seven trained interviewers able to communicate in local languages to elicit information regarding socio-demography, challenges with accessing RMNCH services at the facilities and client satisfaction with these services since the COVID-19 epidemic in Lagos state. The client satisfaction items (S1 Questionnaire) were adapted from a validated tool to measure client-perceived quality of maternity services [18]. All questions were imputed in a form on the KoBoToolbox app (Harvard Humanitarian Initiative, Cambridge, Massachusetts, USA), which was the tool for data collection using smart phones. The instrument was pretested among 20 women of reproductive age in the Lagos environs and necessary adjustments were made to suit the local context. An item on access to credit was removed because it is not applicable to Lagos state health facilities. The research assistants observed strict COVID-19 safety rules such as use of face masks and encouragement of respondents to do same, hand hygiene and data collection in well-ventilated rooms and open spaces.

## Outcome measures

The client satisfaction scales had 19 items in total and assessed satisfaction across four sub-scales: health care delivery, health facility, interpersonal aspects of care and access to services [18, 19]. The options and corresponding scores for these items were: not at all satisfied (score of 1), somewhat satisfied (score of 2) and completely satisfied (score of 3). Those that were not sure of the satisfaction of the specified item or for which the item did not apply were excluded from the analysis for that particular measure. Cronbach's alpha was used to measure the internal consistency of the scale. The health care delivery sub-scale had 7 items with Cronbach alpha value of 0.72 and a possible range of scores between 1 to 21. The health facility sub-scale had 4 items with Cronbach alpha value of 0.73 and a possible range of 1 to 12. The sub-scale for interpersonal aspects of care had 6 items with a Cronbach alpha value of 0.80 and a possible range of 1 to 18. The access to services sub-scale had just 2 items which is less than the minimum of 3 items to calculate Cronbach alpha; it had a possible range of 1 to 6. Overall, the client satisfaction scale had a Cronbach alpha value of 0.86 with a range of 1 to 57.

## Data analysis

Univariate, bivariate and multivariate analyses were used to assess the relationship between the outcome variables and explanatory variables. At the univariate level, we estimated proportions for categorical variables and means and standard deviations for continuous variables. In bivariate analysis, we used one-way analysis of variance to determine statistically significant differences in mean scores across the three levels of care and the Bonferroni Procedure as a post-hoc test. Also, Student's t-test was used to compare the mean client satisfaction scores across patients' personal characteristics. This was followed by multiple linear regression to examine the association between personal characteristics and clients' satisfaction. All independent variables in bivariate analyses with $p$-value $<0.25$ were included in the regression model and beta coefficient and 95% CI were computed for each predictor variable. The results were assessed to be significant at p-value $<0.05$. Data was analysed using STATA version SE15.1 (StataCorp, College Station, Texas, USA).

## Ethics

Ethical approval for this study was obtained from the Health Research and Ethics Committee of Lagos University Teaching Hospital (LUTHHREC/EREV/0620/64). Social approval was

obtained from the Lagos State Ministry of Health and permission to access the facilities was obtained from the Lagos State Health Services Commission and the Lagos State Primary Health Care Board. A waiver of signed consent was obtained from the ethics committee since the research presented minimal risk of harm to participants. Instead, verbal informed consent was obtained from all participants before any interview and documented by the research assistants on the data collection app. Their confidentiality was maintained by not using identifiers in the consent and data collection process.

## Results

### Characteristics of respondents

In all, 1,241 women participated in the study. Their ages ranged from 17–49 years with median age (IQR) of 31 years (28–36). Almost all (93.31%) were married and most had secondary education (50.28%), were employed (81.71%) and perceived their health to be good (50.68%) [Table 1].

### RMNCH services received by beneficiaries

The services most frequently received since the COVID-19 outbreak were childhood immunization (42.02% or 521), treatment of childhood illnesses (27.48% or 341), antenatal care (24.50% or 304) and postnatal care (18.69% or 232) [Fig 1].

**Table 1. Socio-demographic characteristics of clients of RMNCH services during the COVID-19 outbreak in Lagos (N = 1241).**

| Variables | Categories | n (%) | 95% CI |
|---|---|---|---|
| Age group (years) | < 21 | 37 (2.98) | 2.17–4.09 |
| | 21–30 | 526 (42.39) | 39.66–45.16 |
| | 31–40 | 582 (46.90) | 44.13–49.68 |
| | Over 40 | 96 (7.74) | 6.37–9..36 |
| Marital status | Married | 1158 (93.31) | 91.77–94.58 |
| | Single/separated/widowed | 83 (6.69) | 5.42–8.22 |
| Education | No formal | 66 (5.32) | 4.20–6.72 |
| | Primary | 161 (12.97) | 11.21–14.96 |
| | Secondary | 624 (50.28) | 47.50–53.06 |
| | Post-secondary | 390 (31.43) | 28.9–34.1 |
| Religion | Christianity | 799 (64.38) | 61.67–67.00 |
| | Islam | 431 (34.73) | 32.13–37.43 |
| | Traditional | 11 (0.89) | 0.49–1.59 |
| Ethnicity | Yoruba | 700 (56.41) | 53.63–59.15 |
| | Igbo | 339 (27.32) | 24.91–29.87 |
| | Hausa | 65 (5.24) | 4.13–6.63 |
| | Others | 137 (11.04) | 9.41–12.91 |
| Employment status | Employed | 1014 (81.71) | 79.46–83.76 |
| | Unemployed | 227 (18.29) | 16.24–20.54 |
| Facility in which care was received | Primary health care centre | 424 (34.17) | 31.58–36.85 |
| | General hospital | 412 (33.20) | 30.63–35.87 |
| | Teaching hospital | 405 (32.63) | 30.08–35.30 |
| Self-rated state of health | Bad | 31 (2.50) | 1.76–3.53 |
| | Moderate | 283 (22.80) | 20.55–25.22 |
| | Good | 629 (50.68) | 47.90–53.46 |
| | Very good | 298 (24.01) | 21.72–26.47 |

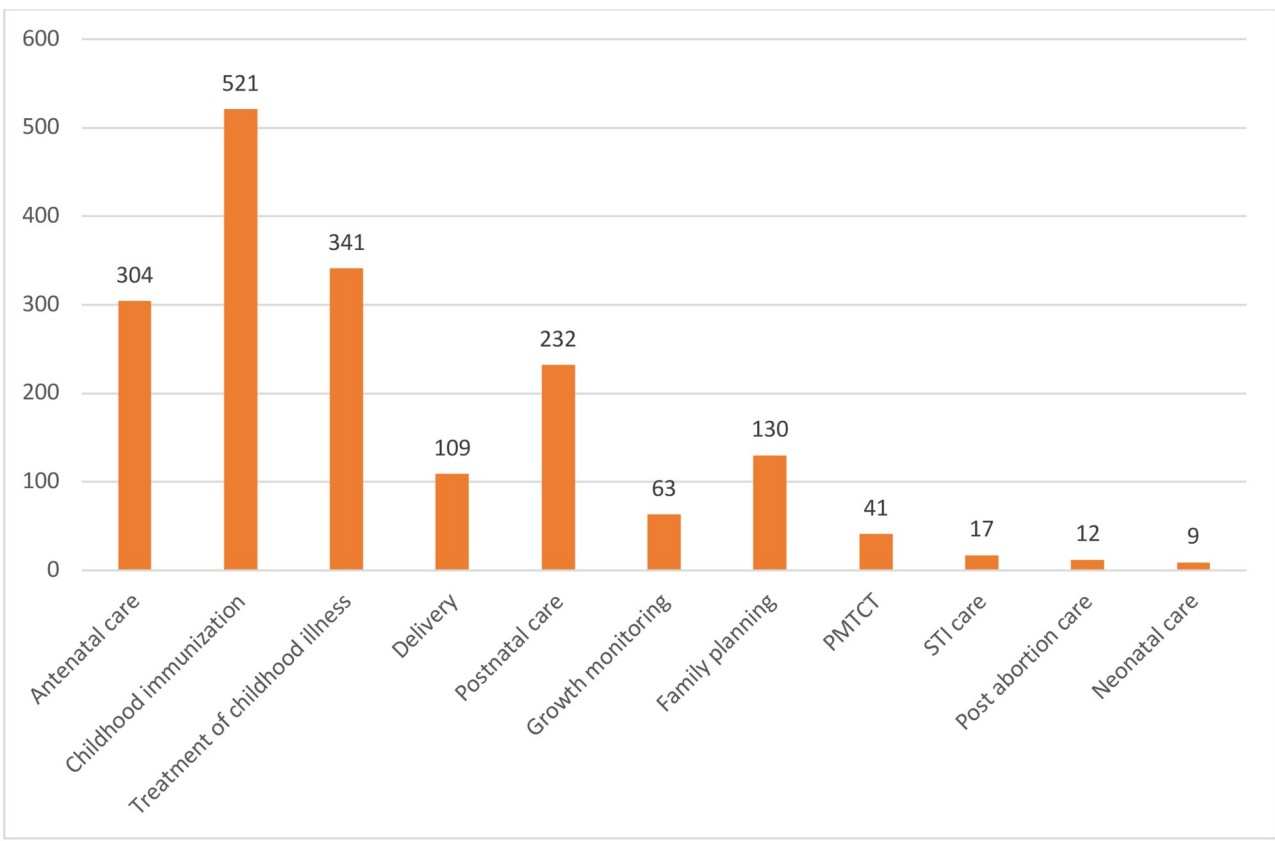

**Fig 1. RMNCH services received during the COVID-19 outbreak in Lagos.**

## Challenges in accessing RMNCH services

Seven hundred and one respondents (56.49%) had no prior challenge accessing RMNCH services since the COVID-19 outbreak. However, the remaining 540 respondents (43.51%) reported a least one challenge with accessing RMNCH services. Close to a third (31.91% or 396) could not access service because they could not leave their houses during the lockdown and 225 (18.13%) could not access service because there was no transportation during the lockdown [Fig 2]. Thirty-six respondents (2.90%) mentioned other challenges such as high cost of transportation, fear of contracting COVID-19/patients with COVID-19 receiving care in the facility and the mandatory use of facemasks at the facility.

## Clients' satisfaction with RMNCH services

The mean clients' satisfaction score among the respondents was 43.25 (SD: 6.28) out of a possible score of 57 (75.88%). Regarding the scores for the sub-scales per facility type, there were statistically significant differences in the mean clients' satisfaction scores for diagnostic skills ($p<0.001$), recovery of patients ($p = 0.012$), monitoring of patient's recovery ($p<0.001$), fee for provided service ($p<0.001$), adequacy of medical equipment ($p = 0.002$), respect for patients ($p = 0.001$), honesty ($p = 0.005$), time spent to explain health status ($p = 0.003$), time devoted to patient ($p = 0.003$), distance to commute to facility ($p<0.001$) and ease of obtaining drugs ($p<0.001$) [Table 2].

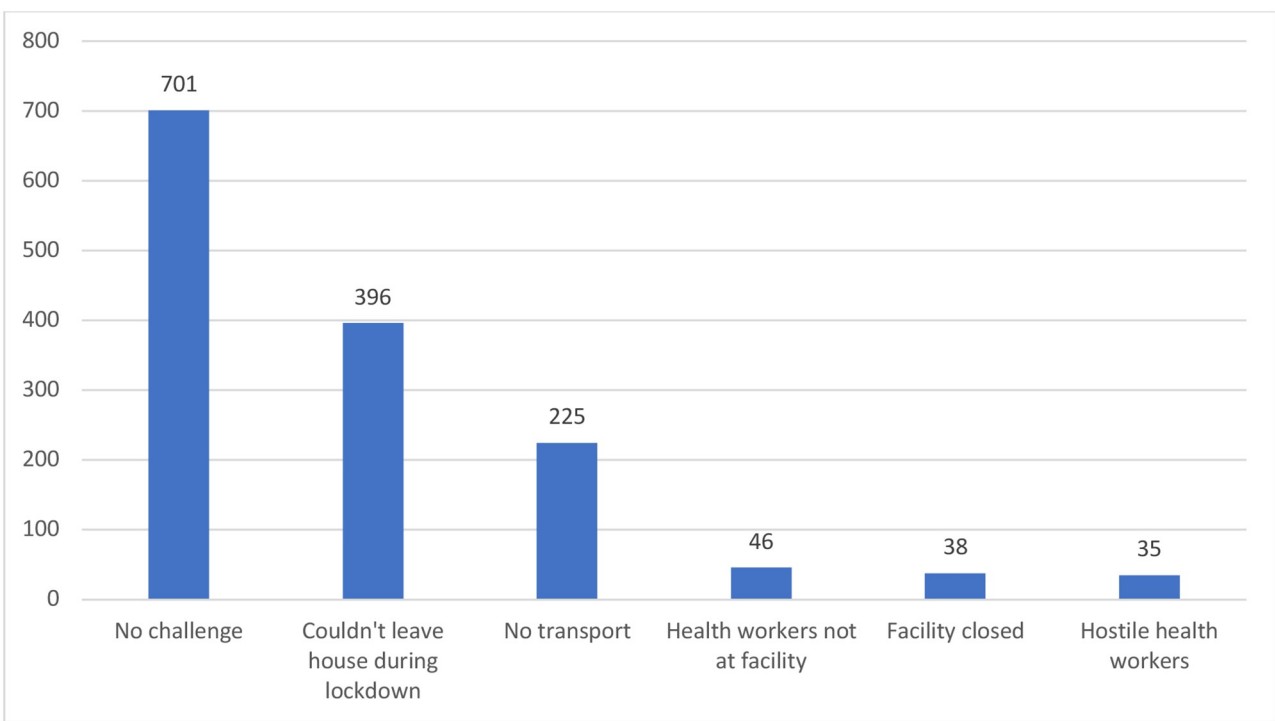

**Fig 2. Challenges in accessing RMNCH services since the COVID-19 outbreak in Lagos.**

Under the health care delivery sub-scale, the post-hoc tests revealed that mean satisfaction score for diagnostic skills was higher in teaching hospitals than PHCs ($p<0.001$), higher in general hospitals than PHCs ($p = 0.037$), not different between the general hospitals and teaching hospitals ($p = 0.326$). Regarding recovery of patients, the mean satisfaction scores was higher for the teaching hospitals than the PHCs ($p = 0.011$) and not different between the PHCs and general hospitals ($p = 1.000$) or between the general hospitals and teaching hospitals ($p = 0.125$). For monitoring of patient's recovery, the mean satisfaction score was higher for the teaching hospitals than the general hospitals ($p = 0.002$), higher for the teaching hospitals than the PHCs ($p = 0.001$) and not different between the PHCs and general hospitals ($p = 1.000$). For fee for provided service, the mean satisfaction score was higher in general hospitals than teaching hospitals ($p<0.001$), higher in PHCs than general hospitals ($p<0.001$) and also higher in PHCs than teaching hospitals ($p<0.001$).

Regarding the health facility sub-scale, the post-hoc test revealed that for adequacy of medical equipment, the mean satisfaction score was higher in general hospitals than PHCs ($p = 0.001$) but not different between PHCs and teaching hospitals ($p = 0.420$) or between general hospitals and teaching hospitals ($p = 0.130$).

For interpersonal aspect of care sub-scale, the mean satisfaction score for compassion for patients was higher in the teaching hospitals than the PHCs ($p = 0.010$) but there was no difference between PHCs and general hospitals ($p = 0.273$) or between general hospitals and teaching hospitals ($p = 0.652$). The mean satisfaction score for respect for patients was higher in teaching hospitals than general hospitals ($p<0.001$), higher in PHCs than general hospitals ($p = 0.001$) but not different between PHCs and teaching hospitals ($p = 1.000$). For honesty, the satisfaction score was higher in the teaching hospitals than general hospitals ($p = 0.003$)

**Table 2. Clients' satisfaction with RMNCH services per facility type during the COVID-19 outbreak in Lagos state.**

| Sub-scales | Primary health care centre | | General hospital | | Teaching hospital | | *p*-value |
|---|---|---|---|---|---|---|---|
| | Mean | 95% CI | Mean | 95% CI | Mean | 95% CI | |
| **Health care delivery** | 15.06 | 14.66–15.47 | 15.57 | 15.26–15.88 | 15.35 | 15.01–15.68 | 0.131 |
| Clinical examination | 2.50 | 2.44–2.55 | 2.56 | 2.51–2.61 | 2.58 | 2.53–2.63 | 0.052 |
| Diagnostic skills | 2.45 | 2.39–2.48 | 2.54 | 2.49–2.60 | 2.61 | 2.56–2.65 | **<0.001** |
| Prescription of drugs | 2.50 | 2.44–2.55 | 2.53 | 2.47–2.58 | 2.54 | 2.49–2.59 | 0.480 |
| Quality of dispensed drugs | 2.54 | 2.48–2.59 | 2.49 | 2.44–2.54 | 2.58 | 2.53–2.63 | 0.070 |
| Recovery of patient | 2.40 | 2.34–2.46 | 2.44 | 2.38–2.49 | 2.52 | 2.46–2.58 | **0.012** |
| Monitoring of patient's recovery | 2.31 | 2.25–2.37 | 2.33 | 2.27–2.39 | 2.47 | 2.42–2.53 | **<0.001** |
| Fee for provided service | 2.63 | 2.57–2.68 | 2.25 | 2.19–2.32 | 1.77 | 1.69–1.85 | **<0.001** |
| **Health Facility** | 9.13 | 8.95–9.30 | 9.33 | 9.15–9.50 | 9.40 | 9.21–9.59 | 0.093 |
| Adequacy of medical equipment | 2.30 | 2.24–2.37 | 2.45 | 2.39–2.51 | 2.37 | 2.31–2.42 | **0.002** |
| Adequacy of consulting rooms | 2.41 | 2.35–2.47 | 2.43 | 2.37–2.49 | 2.37 | 2.31–2.43 | 0.445 |
| Adequacy of staffing | 2.38 | 2.33–2.44 | 2.41 | 2.36–2.47 | 2.34 | 2.28–2.40 | 0.188 |
| Adequacy of health workers for RMNCH | 2.41 | 2.36–2.47 | 2.33 | 2.27–2.38 | 2.40 | 2.34–2.45 | 0.050 |
| **Interpersonal aspects of care** | 14.21 | 13.96–14.45 | 14.03 | 13.81–14.26 | 15.07 | 14.83–15.31 | **<0.001** |
| Compassion for patients | 2.40 | 2.34–2.45 | 2.46 | 2.41–2.52 | 2.51 | 2.46–2.56 | **0.013** |
| Respect for patients | 2.44 | 2.39–2.49 | 2.30 | 2.25–2.36 | 2.45 | 2.40–2.50 | **0.001** |
| Openness to patients | 2.48 | 2.42–2.53 | 2.51 | 2.46–2.56 | 2.54 | 2.49–2.59 | 0.212 |
| Honesty | 2.52 | 2.46–2.57 | 2.46 | 2.41–2.51 | 2.58 | 2.54–2.63 | **0.005** |
| Time spent to explain health status of the woman or child | 2.40 | 2.34–2.45 | 2.33 | 2.28–2.39 | 2.50 | 2.41–2.51 | **0.003** |
| Time devoted to patient | 2.43 | 2.37–2.49 | 2.40 | 2.35–2.46 | 2.53 | 2.48–2.58 | **0.003** |
| **Access to services** | 4.39 | 4.28–4.50 | 4.50 | 4.39–4.60 | 3.74 | 3.64–3.84 | **<0.001** |
| Distance to commute to facility | 2.17 | 2.10–2.23 | 2.20 | 2.14–2.26 | 1.72 | 1.64–1.79 | **<0.001** |
| Ease of obtaining drugs | 2.43 | 2.37–2.49 | 2.39 | 2.33–2.45 | 2.19 | 2.13–2.24 | **<0.001** |

CI: Confidence Interval

but not different between PHCs and teaching hospitals (*p* = 0.262) or between PHCs and general hospitals (*p* = 0.344). For time spent to explain health status, the mean score was higher in teaching hospitals than general hospitals (*p* = 0.002) but not different between PHCs and teaching hospitals (*p* = 0.271) or between PHCs and general hospitals (*p* = 0.265). For the time devoted to patients, the mean score was higher for teaching hospitals than general hospitals (*p* = 0.004), higher in teaching hospitals than PHCs (*p* = 0.036) but not different between PHCs and general hospitals (*p* = 1.000).

In the access to services sub-scale, the mean score for distance to commute to facility was higher in general hospitals than teaching hospitals (*p*<0.001), higher in PHCs than teaching hospitals (*p*<0.001) but not different between PHCs and general hospitals (*p* = 1.000). For ease of obtaining drugs, the mean score was higher in general hospitals than teaching hospitals (*p*<0.001), higher in PHCs than teaching hospitals (*p*<0.001) but not different between PHCs and general hospitals (*p* = 0.918).

## Factors associated with clients' satisfaction with RMNCH services

Table 3 shows the bivariate analysis examining the association between personal characteristics of respondents and clients' satisfaction. Respondents above 30 years of age had significantly higher mean satisfaction scores than those aged 30 years and below (*p*<0.001).

**Table 3. Bivariate analysis of personal characteristics and clients' satisfaction with RMNCH services during the COVID-19 outbreak in Lagos.**

| Variables | Categories | Mean client satisfaction score (SD) | p-value |
|---|---|---|---|
| Age group | 30 and below | 42.29 (6.21) | <0.001 |
| | Above 30 | 44.04 (6.24) | |
| Marital status | Married | 43.28 (6.34) | 0.420 |
| | Not married | 42.71 (5.40) | |
| Education | Less than post-secondary | 43.30 (6.16) | 0.155 |
| | Post-secondary | 43.14 (6.55) | |
| Tribe | Yoruba | 43.38 (6.30) | 0.414 |
| | Others | 43.08 (6.26) | |
| Religion | Christianity | 43.04 (6.44) | 0.122 |
| | Others | 43.62 (6.03) | |
| Employment status | Employed | 43.20 (6.30) | 0.540 |
| | Unemployed | 43.48 (6.28) | |
| Facility of care | Primary health care centre | 42.78 (6.80) | 0.460 |
| | General hospital | 43.42 (5.70) | |
| | Teaching hospital | 43.56 (6.28) | |
| Self-rated state of health | Moderate-bad | 43.42 (6.06) | 0.583 |
| | Good-very good | 43.19 (6.36) | |

In the multivariable linear regression model that assessed the predictors of clients' satisfaction, being over 30 years of age was significantly associated with an increased clients' satisfaction score (ß = 1.80, 95%CI: 1.10–2.50) [Table 4].

## Discussion

The objective of this study was to understand the challenges faced by women who used RMNCH services in Lagos, Nigeria, their satisfaction with care received during the COVID-19 pandemic and the factors associated with their satisfaction. A third of women could not access

**Table 4. Linear regression model showing predictors of clients' satisfaction with RMNCH services during the COVID-19 outbreak in Lagos.**

| Variables | ß [95%CI] | p-value |
|---|---|---|
| **Age group** | | |
| 30 and below (Ref.) | | |
| Above 30 | 1.80 [1.10–2.50] | <0.001 |
| **Education** | | |
| Less than post-secondary (Ref.) | | |
| Post-secondary | - 0.13 [-0.88–0.63] | 0.743 |
| **Religion** | | |
| Christianity (Ref.) | 0.69 [-0.04–1.43] | 0.064 |
| Other | | |
| *Intercept* | *9.35 [3, 1237); <0.001* | |
| *N-size* | *1241* | |
| *R-square* | *0.022* | |
| *Adj. R-square* | *0.020* | |
| *RMSE* | *6.221* | |

Ref: Reference categories; ß: Beta coefficient; CI: Confidence interval; RMSE: root mean square error

RMNCH services during the pandemic, with the most prevalent reasons being that they could not leave their houses during the lockdown and no transport available. The lockdown and travel restrictions put in place by the different state governments in the height of the pandemic were reasons that were given for this barrier in access. This limited access to RMNCH services was a shared experience in many LMICs during this period [13, 14, 20]. However, in our study, to a limited extent, barriers such as health workers not being at their workplace and health facilities being closed were reported by about three percent of women. While it is concerning that some women were essentially 'locked out' of the health system during the lockdown due to service reduction, it supports reports of skilled health personnel that they were still working during the lockdown [14]. In particular, the Lagos state government made use of innovative practices to ensure RMNCH services were still continuing during the lockdown [21]. Examples of such were the provision of free antenatal and delivery (including surgical) services as well as the provision of free drugs and laboratory services at comprehensive PHCs and general hospitals. Some of these actions were already being done before the pandemic including free registration and free antenatal care, however, delivery and postnatal care were not free at point-of-use for women [22]. Anecdotal evidence at the local government level revealed that further support was provided for the PHCs by ensuring that skilled health personnel were picked up from their homes and provided with food and in some instances, accommodation, to keep RMNCH facilities running.

The overall satisfaction score of over 70% in our study was similar to another study that used the same sub-scales to assess Nepalese women's satisfaction with maternity services [23]. Regarding the scores for the sub-scales per facility type, there were statistically significant differences in the mean clients' satisfaction scores for diagnostic skills, recovery of patient, monitoring of patient's recovery, fee for provided service, adequacy of medical equipment, respect for patients, honesty, time spent to explain health status, time devoted to patient, distance to commute to facility and ease of obtaining drugs ($p < 0.05$). A previous review classified factors into structural (such as good physical environment, facility cleanliness, and availability of adequate skilled health personnel, medicines and supplies), process (such as privacy, respect, timeliness, perceived provider competency and compassion) and outcome-related factors [24]. Structural factors are not expected to be specifically related to COVID-19 and certainly not vary because of the pandemic. For example, it makes sense that satisfaction scores around diagnostic skills and monitoring of patient's recovery were generally higher in the teaching and general hospitals. This is probably because they have higher capacity for these services. Similarly, satisfaction scores relating to the fees being paid for care were lowest in teaching hospitals. Again, this is expected, as higher cost of care utilisation has been reported in such facilities due to the relatively wider array of specialist care, hi-tech medical equipment and overheads. For access to services, this would not also have varied pre- and post- the emergence of COVID-19, as the health facilities have always been in the same location and there were no additional facilities built during the period.

However, with interpersonal aspects of care, these factors could have been influenced by the COVID-19 pandemic. A global survey of health workers on the frontline revealed that 90% had experienced somewhat or substantially higher levels of stress [14]. Pre-pandemic, these factors have been widely reported as being as particularly important for women in LMICs [24, 25]. Our study shows that satisfaction scores were statistically significantly lower in the PHCs and general hospitals compared to teaching hospitals. While to the best of our knowledge, no comparative satisfaction study has been conducted for RMNCH services across the three-tiers in Lagos, Nigeria, evidence shows that women in Nigeria value RMNCH care at the PHC level for its proximity to their residence, good quality care and availability of a provider [26, 27]. In terms of the relative higher satisfaction in the tertiary level hospitals compared to PHCs, this is

certainly a major concern and warrants further investigation, especially as a few studies in the region have raised concern with interpersonal aspects of care at hospital level pre-pandemic [17, 28]. In one of those studies, only 12% of women rated interpersonal aspects of care as 'good' in a specialist public hospital, with 70% rating services as 'fair' [17]. It is also possible that these relatively lower scores are also due to the composition of women who typically use PHCs.

However, when scores of interpersonal aspects of care were disaggregated to sub-scales, there were lower scores for compassion, respect for patients, honesty and time spent to explain health status at primary and secondary levels. While all health workers, irrespective of tier, have reported some strain on their capacity to provide RMNCH services during the pandemic, it might be the case that health personnel in these facilities feel even less sufficiently equipped to give time to explain health status to women, which clients may interpret as dishonest. Their social distancing to provide safe care to patients while also quickly trying to get them out of the clinic may have appeared as disrespect. A pointer to this might be the fact that many of the donations for PPEs to health workers were made directly to teaching hospitals [29, 30]. One other plausible explanation may be dearth in respectful care training in the context of COVID-19. As per a survey conducted between March and July 2020, more than 75% of RMNH providers in the state who work at the PHC level, had received some COVID-19 training to help them provide RMNH services (higher than providers in secondary and tertiary hospitals) [31]. However, while this training focused on technical provision of care, there may have been a missed opportunity in building capacity of providers on the non-technical aspects of care, which women deem particularly important [28, 32].

Age was a particularly significant predictor of satisfaction with RMNCH services, with women 30 years and below rating the services lower than older women. Though previous studies conducted in Lagos have shown that factors such as age, marital status, occupation, income, and type of facility are significant predictors of satisfaction with health care [27, 33], we only found age to be a significant factor. Indeed, it is not particularly surprising that no other factors were significant in our study compared to others, as the impact of COVID-19 on care service has been global, irrespective of socio-economic status or marriage. However, for age, we find that this might be related more with the generational gap and higher expectations of millennials for higher quality healthcare [34].

Our findings have huge implication for practice and policy. For policy, the criticality and risk involved in providing MNH care in the fragility of the Nigerian health system during the pandemic have already been highlighted in the literature [35]. While care provision has been affected grossly during the pandemic, our study has shown that there is greater concern at the tier of care that would be most important for communities–PHC. Focusing on PHC is particularly important in the context of Nigeria, being closer-to-the community. Also, PHC and use of lower-cadre health workers has been argued to be the golden goose needed to address care provision during the pandemic [36, 37]. Incorporating tailored respectful maternity care [38] as part of on-going COVID-19 trainings will make a significant difference at all levels, more so for PHC workers. For clinical practice, addressing fear of COVID-19 in patients is one constant factor associated with satisfaction with care, as has been shown in a study conducted with mothers in Pakistan during this pandemic [39].

## Strengths and limitations

There are some key strengths worth highlighting regarding our study. This being the first large-sample, multi-facility study conducted in the epicentre of COVID-19 in Nigeria, we were able to establish valid inferences that can be helpful for evidence-based decision-making in the

middle of the ongoing pandemic. In addition, we used a validated tool that has been used in similar LMIC settings [18, 19]. Also, the fact that we conducted the study with clients just after they have received services ensured the recency of their assessment of the service received. However, our findings need to be interpreted keeping in mind certain limitations. First, it is possible that there could be some social desirability bias in the responses from clients. Second, this study was conducted in Lagos, the economic nerve centre of Nigeria with more human and financial resources for health. Thus, the findings are not generalizable to other parts of the country. Third, experiences around COVID-19 are fluid and could change radically with changing policies. Hence, our findings should be interpreted with reference to the timing of the study.

## Conclusion

In many LMICs, including Nigeria, huge gains were already being made towards realising universal health coverage and improving maternal and newborn health outcomes. However, COVID-19 has disrupted the normal, necessitating new thinking to protect these gains [40]. Our findings are consistent with previous studies in LMICs, which have reported challenges faced by women in accessing RMNCH services during the COVID-19 lockdown. Although overall satisfaction with care was fairly high, there should be an increased focus on the needs of women of all ages, their newborns and children in LMICs during COVID-19 [41]. This should include providing tailored COVID-19 sensitive inter-personal care to them at all levels of care.

## Supporting information

**S1 Questionnaire. Original survey questionnaire.**
(DOCX)

**S1 Dataset. Dataset of 1241 clients of RMNCH services during the COVID-19 outbreak in Lagos.**
(XLSX)

## Acknowledgments

We are indebted to Dr Adesola Pitan (Deputy Director of Medical Services, Lagos State Health Service Commission) and the facility managers of the twenty-two health facilities used in this study, for their immense support during data collection.

## Author Contributions

**Conceptualization:** Mobolanle Balogun, Aduragbemi Banke-Thomas, Adekemi Sekoni, Godfred O. Boateng, Bosede B. Afolabi, Folasade Ogunsola.

**Data curation:** Mobolanle Balogun.

**Formal analysis:** Mobolanle Balogun.

**Investigation:** Mobolanle Balogun, Aduragbemi Banke-Thomas.

**Methodology:** Mobolanle Balogun, Aduragbemi Banke-Thomas, Adekemi Sekoni, Godfred O. Boateng, Victoria Yesufu, Ololade Wright, Osinachi Ubani, Akin Abayomi, Bosede B. Afolabi, Folasade Ogunsola.

**Project administration:** Mobolanle Balogun, Adekemi Sekoni, Godfred O. Boateng, Victoria Yesufu, Ololade Wright, Osinachi Ubani.

**Resources:** Mobolanle Balogun, Aduragbemi Banke-Thomas, Bosede B. Afolabi, Folasade Ogunsola.

**Supervision:** Akin Abayomi, Bosede B. Afolabi, Folasade Ogunsola.

**Writing – original draft:** Mobolanle Balogun, Aduragbemi Banke-Thomas.

**Writing – review & editing:** Mobolanle Balogun, Aduragbemi Banke-Thomas, Adekemi Sekoni, Godfred O. Boateng, Victoria Yesufu, Ololade Wright, Osinachi Ubani, Akin Abayomi, Bosede B. Afolabi, Folasade Ogunsola.

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
