## [Decision Letter · Decision Letter 0]

30 Mar 2021

PONE-D-21-02016

Challenges in access and satisfaction with reproductive, maternal, newborn and child health services in Nigeria during the COVID-19 pandemic: A cross-sectional survey

PLOS ONE

Dear Dr. Balogun,

Thank you for submitting your manuscript to PLOS ONE. After careful consideration, we feel that it has merit but does not fully meet PLOS ONE’s publication criteria as it currently stands. Therefore, we invite you to submit a revised version of the manuscript that addresses the points raised during the review process.

We look forward to receiving your revised manuscript.

Kind regards,

Mary Hamer Hodges, MBBS MRCP DSc

Academic Editor

PLOS ONE

Journal Requirements:

2. Please include additional information regarding the survey or questionnaire used in the study and ensure that you have provided sufficient details that others could replicate the analyses. For instance, if you developed a questionnaire as part of this study and it is not under a copyright more restrictive than CC-BY, please include a copy, in both the original language and English, as Supporting Information. Moreover, please include more details on how the questionnaire was pre-tested, and whether it was validated.

3. Please provide additional details regarding participant consent. In the ethics statement in the Methods and online submission information, please ensure that you have specified why  verbal consent was chosen, and how it was  documented and witnessed.

Reviewers' comments:

Reviewer's Responses to Questions

**Comments to the Author**

1. Is the manuscript technically sound, and do the data support the conclusions?

Reviewer #1: Yes

Reviewer #2: Yes

2. Has the statistical analysis been performed appropriately and rigorously? 

Reviewer #1: Yes

Reviewer #2: Yes

3. Have the authors made all data underlying the findings in their manuscript fully available?

Reviewer #1: Yes

Reviewer #2: Yes

4. Is the manuscript presented in an intelligible fashion and written in standard English?

Reviewer #1: Yes

Reviewer #2: Yes

5. Review Comments to the Author

Reviewer #1: I read with interest the paper by Balogun et al that assessed the challenges faced by women who used RMNCH services in Nigeria’s epicentre and women’s satisfaction with care received during the COVID-19 pandemic. The methodology and the results of the paper are in in general appropriate for answering the research question. I have some comments for the scientific writing. Comments are attached below:

•Objective— The current objectives of the article cannot capture some of the main results in the article. For example, the analysis of the relationship between personal characteristics and “client satisfaction” is a main result reported. We cannot see from the objective that the authors intended to answer such question

•Method —the authors are recommended to define RMNCH services (line 117) in this study and give examples.

•Results —Authors need to make the digits of numbers consistent as well. For example, table 1, 3 have one decimal place for the numbers, whilst table 2 and 4 have two decimal places. The p values also need to be consistent in digits.

•Conclusion-The authors concluded that the overall satisfaction with care was fairly high. This is a cross-sectional study, so before and after comparison is not possible. The authors need to consider using benchmarks to support their interpretation. As the authors stated that they used a “validated tool” in this study, evidence from previous studies using the same tool could be brought in to make comparison.

•Authors should try to use consistent terminology throughout the document. Is there a difference between patient satisfaction (page 3) and client satisfaction (introduction and main text)? if not, sticking to one term is recommended.

Reviewer #2: This is a well written and interesting study. It is very clearly presented and has thorough and logical discussion and conclusion sections. It will certainly add to the knowledge about services provided during Covid-19. There are a few minor edits suggested in the attachment. The limitation of the study being conducted in Lagos, which has generally higher standards of delivery and health outcomes than other parts of Nigeria, is well noted.

6. PLOS authors have the option to publish the peer review history of their article (what does this mean?). If published, this will include your full peer review and any attached files.

Reviewer #1: No

Reviewer #2: **Yes: **Paula Quigley

---

## [Author Response · Author response to Decision Letter 0]

6 Apr 2021

REVIEWER 1

Comment #1: 

Objective— The current objectives of the article cannot capture some of the main results in the article. For example, the analysis of the relationship between personal characteristics and “client satisfaction” is a main result reported. We cannot see from the objective that the authors intended to answer such question.

Response: Thank you for the careful read of our paper and for such insightful comments. We have edited the objective in the abstract, introduction and discussion to read “The objective of this study was to understand the challenges faced by women who used RMNCH services in Lagos, Nigeria, their satisfaction with care received during the COVID-19 pandemic and the factors associated with their satisfaction.”

Comment #2: 

Method —the authors are recommended to define RMNCH services (line 117) in this study and give examples.

Response: This has been defined and examples of RMNCH services provided across the levels of care have been stated in lines 119 – 127. 

Comment #3: 

Results —Authors need to make the digits of numbers consistent as well. For example, table 1, 3 have one decimal place for the numbers, whilst table 2 and 4 have two decimal places. The p values also need to be consistent in digits.

Response: Thanks for this comment. We have made all numbers in the results two decimal places. The p- values are consistently three decimal places. 

Comment #4: 

Conclusion-The authors concluded that the overall satisfaction with care was fairly high. This is a cross-sectional study, so before and after comparison is not possible. The authors need to consider using benchmarks to support their interpretation. As the authors stated that they used a “validated tool” in this study, evidence from previous studies using the same tool could be brought in to make comparison.

Response: We have highlighted that the overall satisfaction score was over 70% and compared it to another study that used the same sub-scales in lines 334-335. We used just one other study for comparison because other studies that used the same tool presented scores for the sub-scales but did not present the overall scores. Those other studies mentioned are: 

• Devkota HR, Clarke A, Murray E, Groce N. Do experiences and perceptions about quality of care differ among social groups in Nepal? : A study of maternal healthcare experiences of women with and without disabilities, and Dalit and non-Dalit women. PLoS One. 2017 Dec 19;12(12):e0188554. doi: 10.1371/journal.pone.0188554. PMID: 29261691; PMCID: PMC5736179.

• Erchafo B, Alaro T, Tsega G, Adamu A, Yitbarek K, Siraneh Y, Hailu M, Woldie M. Are we too far from being client centered? PLoS One. 2018 Oct 15;13(10):e0205681. doi: 10.1371/journal.pone.0205681. PMID: 30321212; PMCID: PMC6188795.

Comment #5: 

Authors should try to use consistent terminology throughout the document. Is there a difference between patient satisfaction (page 3) and client satisfaction (introduction and main text)? if not, sticking to one term is recommended.

Response: Thank you for drawing our attention to this inconsistency. We have changed patient satisfaction to client satisfaction in the places where it occurs.

REVIEWER 2

This is a well written and interesting study. It is very clearly presented and has thorough and logical discussion and conclusion sections. It will certainly add to the knowledge about services provided during Covid-19. There are a few minor edits suggested in the attachment. The limitation of the study being conducted in Lagos, which has generally higher standards of delivery and health outcomes than other parts of Nigeria, is well noted.

Response: Thank you for your kind comments. We have included the corrections in the attachment in the revised manuscript.

---

## [Decision Letter · Decision Letter 1]

26 Apr 2021

Challenges in access and satisfaction with reproductive, maternal, newborn and child health services in Nigeria during the COVID-19 pandemic: A cross-sectional survey

PONE-D-21-02016R1

Dear Dr. %Mobolanle Balogun%,

We’re pleased to inform you that your manuscript has been judged scientifically suitable for publication and will be formally accepted for publication once it meets all outstanding technical requirements.

Kind regards,

Mary Hamer Hodges, MBBS MRCP DSc

Academic Editor

PLOS ONE

Additional Editor Comments (optional):

Reviewers' comments:

Reviewer's Responses to Questions

**Comments to the Author**

1. If the authors have adequately addressed your comments raised in a previous round of review and you feel that this manuscript is now acceptable for publication, you may indicate that here to bypass the “Comments to the Author” section, enter your conflict of interest statement in the “Confidential to Editor” section, and submit your "Accept" recommendation.

Reviewer #1: All comments have been addressed

Reviewer #2: All comments have been addressed

2. Is the manuscript technically sound, and do the data support the conclusions?

Reviewer #1: Yes

Reviewer #2: (No Response)

3. Has the statistical analysis been performed appropriately and rigorously? 

Reviewer #1: Yes

Reviewer #2: (No Response)

4. Have the authors made all data underlying the findings in their manuscript fully available?

Reviewer #1: Yes

Reviewer #2: (No Response)

5. Is the manuscript presented in an intelligible fashion and written in standard English?

Reviewer #1: Yes

Reviewer #2: (No Response)

6. Review Comments to the Author

Reviewer #1: (No Response)

Reviewer #2: (No Response)

7. PLOS authors have the option to publish the peer review history of their article (what does this mean?). If published, this will include your full peer review and any attached files.

Reviewer #1: No

Reviewer #2: **Yes: **Paula Quigley

---

## [Editor Report · Acceptance letter]

30 Apr 2021

PONE-D-21-02016R1 

Challenges in access and satisfaction with reproductive, maternal, newborn and child health services in Nigeria during the COVID-19 pandemic: A cross-sectional survey 

Dear Dr. Balogun:

I'm pleased to inform you that your manuscript has been deemed suitable for publication in PLOS ONE. Congratulations! Your manuscript is now with our production department. 

Kind regards, 

on behalf of

Dr. Mary Hamer Hodges 

Academic Editor

PLOS ONE